# FKBP5 rs4713916: A Potential Genetic Predictor of Interindividual Different Response to Inhaled Corticosteroids in Patients with Chronic Obstructive Pulmonary Disease in a Real-Life Setting

**DOI:** 10.3390/ijms20082024

**Published:** 2019-04-24

**Authors:** Patrizia Russo, Carlo Tomino, Alessia Santoro, Giulia Prinzi, Stefania Proietti, Aliaksei Kisialiou, Vittorio Cardaci, Massimo Fini, Mauro Magnani, Francesco Collacchi, Mauro Provinciali, Robertina Giacconi, Stefano Bonassi, Marco Malavolta

**Affiliations:** 1Clinical and Molecular Epidemiology, IRCCS San Raffaele Pisana, Via di Valcannuta, 247 00166 Rome, Italy; alessiasantoro92@gmail.com (A.S.); giulia.prinzi@gmail.com (G.P.); alesseus@gmail.com (A.K.); Stefano.bonassi@sanraffaele.it (S.B.); 2Scientific Direction, IRCCS San Raffaele Pisana, 0166 Rome, Italy; carlo.tomino@sanraffaele.it (C.T.); stefania.proietti@sanraffaele.it (S.P.); massimo.fini@sanraffaele.it (M.F.); 3Pulmonary Rehabilitation, IRCCS San Raffaele Pisana, 00166 Rome, Italy; vittorio.cardaci@sanraffaele.it; 4Department of Biomolecular Science-Section of Biotechnology, University of Urbino “Carlo Bo”, 61032 Fano, Italy; mauro.magnani@uniurb.it; 5Diatheva Srl, 61030 Pesaro e Urbino, Italy; f.collacchi@diatheva.com; 6Advanced Technology Center for Aging Research, Scientific Technological Area, IRCCS INRCA, 60124 Ancona, Italy; m.provinciali@inrca.it (M.P.); r.giacconi@inrca.it (R.G.); M.MALAVOLTA@inrca.it (M.M.)

**Keywords:** COPD, inhaled corticosteroid, lung function, rehabilomics, rs37972, rs471396

## Abstract

*Background*: Chronic obstructive pulmonary disease (COPD) is a common, preventable, and manageable lung disease characterized by large heterogeneity in disease presentation and grades impairment. Inhaled corticosteroids (ICS) are commonly used to manage COPD/COPD-exacerbation. The patient’s response is characterized by interindividual variability without disease progression/survival modification. *Objectives*: We hypothesize that a therapeutic intervention may be more effective if single nucleotide polymorphisms (SNPs) are investigated. *Methods*: In 71 COPD patients under pulmonary rehabilitation, a small number of powerful SNPs, selected according to current literature, were analyzed; namely the glucocorticoid receptor gene NR3C1 (rs6190/rs6189/rs41423247), the glucocorticoid-induced transcript 1 gene (GLCCI1 rs37972), and the related co-chaperone FKBP5 gene (rs4713916). MDR1 rs2032582 was also evaluated. Lung function outcomes were assessed. *Results*: A significant association with functional outcomes, namely FEV_1_ (forced expiration volume/one second) and 6MWD (six-minutes walking distance), was found for rs4713916 and weakly for rs37972. The genotype rs4713916(GA) and, in a lesser extent, the genotype rs37972(TT), were more favorable than the wild-type. *Conclusions*: Our study supports a possible picture of pharmacogenomic control for COPD intervention. rs4713916 and, possibly, rs37972 may be useful predictors of clinical outcome. These results may help to tailor an optimal dose for individual COPD patients based on their genetic makeup.

## 1. Introduction

Until now, medicine was essentially founded on the paradigm of the “evidence-based” for the care of patients [1,2]. Although evidence-based medicine (EBM) represents a significant improvement over the past “intuition-based medicine”, actually its limitations are emergent [3]. Thus, medical practice is based on “clinical practice guidelines”, obtained by treatment of a combination of different patients all sharing a common phenotype (i.e., disease). In applying strictly the clinical practice guidelines, a patient is considered as a member of a group showing similar characteristics, health condition, and stage of progression, and treated as “per usual”. Obviously, this approach normally does not only not consider the heterogeneity of the person but also of that of the disease. Chronic obstructive pulmonary disease (COPD) is characterized by a large heterogeneity in disease presentation and grades of impairment, and is one of the leading causes of morbidity and mortality worldwide. In the European Union, nearly 300,000 people die of COPD every year [4,5]. The data are probably higher since COPD is under-recognized, under-diagnosed, and under-treated [6]. COPD is a widespread, preventable, and manageable lung disease characterized by severe airflow limitation, reduced elastic recoil, and parenchymal tethering that induces dyspnea, symptoms of breathlessness, exercise intolerance, cough, and sputum production [7]. 

The current pharmacological treatment options are limited, mostly symptomatic, and generally do not alter disease progression, with no effect on survival [8]. In this contest pulmonary rehabilitation (PR) is the most effective therapeutic strategy to improve shortness of breath, health status, and exercise tolerance reducing readmissions and mortality [7]. The NIH’s recent research plan on rehabilitation states that part of its translational science goals are to: “(i) Advance the understanding of precision medicine approaches relevant to rehabilitation medicine; and (ii) characterize biomarkers associated with specific injuries, illnesses, or disorders that are prognostic or guide prescription of rehabilitation interventions” [9]. The so-called “Rehabilomics” research framework, initially introduced by Wagner in traumatic brain injury rehabilitation [10], may be applied to COPD, integrating areas of contemporary clinical pneumology, biomarkers, and genetic research. As suggested by Wagner [10], the “Rehabilomics” represents a unique and distinctive model for a foundation of a developing personalized-medicine approaches to rehabilitation. In our previous work we employed a Rehabilomics-like approach to study the heterogeneity in COPD rehabilitation, incorporating clinical and psychological/emotional traits as well as biological (i.e., gender) factors [11]. 

In individuals with persistent symptoms or frequent exacerbations, inhaled corticosteroids (ICS) or systemic corticosteroids are largely used [7]. Clinical response to corticosteroids (CSs) is heterogeneous since inter-individual variability affects the effectiveness of treatment, and consequently many COPD patients show an insufficient or absent response [12].

Different molecular mechanisms of resistance to CSs have been identified, among these, dysregulated expression of GCSs-receptor isoforms is classically involved in the sensitivity/resistance [13]. To be active, CSs shall bind to the glucocorticoids receptor (GR). GR is a ligand-induced transcription factor member of the nuclear receptor superfamily transcribed by a single gene, the NR3C1 [14]. Single-nucleotide polymorphisms (SNPs) in NR3C1 (rs6190, rs6189, bcl2, and rs41423247), in the glucocorticoid-induced transcript 1 gene (GLCCI1 rs37972) or in the related co-chaperone FKBP5 genes (rs4713916), are associated with sensitivity/resistance to CSs [15,16]. Indeed, a strong association exists between the presence of the minor allele A in the rs4713916 and the responsiveness to CSs therapy in Crohn’s disease patients [17].

The minor rs37972 allele (GG) has been initially associated with a decreased response to ICS therapy in asthmatics patients [18,19], as well as in COPD patients [20]. Afterward, discordant results are reported: One study shows no evidence between the minor rs37972 allele and CSs response in patients with COPD [20], whereas a second study reports association between the minor rs37972 allele and a decreased ICS efficacy in Chinese COPD patients [21]. Since, the first study was performed on a non-Hispanic Caucasian population and the second on a Chinese population, a possible ethnically-specific genetic susceptibility difference may be hypothesized.

Additionally, SNPs in the multi-drug resistance gene MDR1 (ABCB1), encoding the drug efflux pump P-glycoprotein 170, have been associated to CSs resistance [22,23].

The aim of this study is to investigate the role of selected SNPs of the NR3C1 (rs6190, rs6189, bcl2, and rs41423247), of the FKBP5 (rs4713916), of the GLCCI1 (rs37972) and of the MDR1 (rs2032582) gene in modulating the effect of CSs on therapeutic endpoints in a group of COPD patients. 

## 2. Results

A total of 71 patients (mean age ± SD: 72.77 ± 8.49 years) with severe COPD, stage 3–4 GOLD, were recruited. The majority of patients, 57.8%, were female. The demographic characteristics, lung functions, and pathological anamnesis at baseline of the study group are reported in Table 1. 

According to this table, patients were moderately overweight [24], did not report major cognitive impairment, had an impaired disease-specific health status, a reduced exercise tolerance, and 32.4% of them were under LTOT. At baseline, in 6MWT, patients walked 96.8 ± 85.3 m, reflecting the severity of their health status and the high variability in the clinical response. 

At the end of the three-week inpatient PR program, all respiratory outcome measures showed significant improvement (Table 2). 

Specifically, the 6MWD improved by 93.56 ± 79.97 m. Importantly, patients classified as GOLD stage 3 (severe COPD) after PR were classified as stage 2 (mild COPD).

### Genotype and Association with Outcomes of Pulmonary Rehabilitation

The allele frequency of SNPs among the four genes investigated, within the COPD patients under study, is shown in Table 3. The allele frequency was consistent with the European frequency for these genes [25,26,27,28,29,30]. However, in our 71 samples the allele AA for rs4713916 was not detected. The frequency of allele A was reported to be around 0.22 [30].

Association between SNPs and response to PR, evaluated as Δ6MWD or ΔFEV_1_, was analyzed. According to the results of 6MWD, 21 patients (29.6%) were considered non-responders (Δ < 30 m) and 50 (70.4%) as responders (Δ > 30 m). Since, among the 50 responders, only two showed a Δ6MWD below 50 m (i.e., 40 m), the analysis was performed on the remaining 48 responders (Δ > 50 m) and on 21 non-responders (Δ < 30 m) to enhance the study feasibility, excluding intermediate subjects. We hypothesized an additive model to reveal a possible association between SNPs, associated to the metabolism of CSs treatment and response to PR in term of 6MWT, since a Mendelian inheritance of this SNPs is still unknown [31].

Table 4 shows the relationships between genotype and allele distribution of SNP among the four genes in COPD patients and response to ICS according to 6MWD. 

rs4713916 (GA) in the FKBP5 gene was nominally associated with an increase in 6MWD and rs37972 (TT) in the GLCCI1 gene showed a tendency to be positive associated with an increase in 6MWD; however, the number of patients carrying TT was too small to be significant. rs6189, rs6190, and rs41423247, in the NR3C1 gene, as well as rs2032582 in the MDR1, were not associated with any increase in 6MWD. The relationship with Δ6MWD and improvement in FEV_1_ for rs4713916 or for rs37972 is shown in Figure 1

Figure 1 shows, clearly, a positive association between the genotype rs4713916 GA and the two outcomes examined. For the genotype rs37972 TT, no clear correlation was observed.

Table 5 shows selected characteristics (statistically significant) of COPD patients stratified by FKBP5 variant rs4713916. 

The GA carriers showed, also, an excellent improvement in lung function, evaluated as SGRQ total score and Borg index of dyspnea. Among available blood test parameters, serum bilirubin is considered one potential “routine” candidate biomarker of COPD [32,33,34]. GA carrier patients showed higher levels of serum bilirubin than GG carriers (0.94 ± 0.55 *versus* 0.58 ± 0.29 mg/dL, *p* = 0.0453, respectively) supporting the hypothesis that these patients respond better to IPR. 

No association with personal smoking/alcohol drinking habits and medical family disease history, including respiratory illness, was observed ().

Logistic regression analysis shows that patients carrying rs4713916 GA had four times more probability to be responders than GG patients (CI = 0.02–1.68, odd ratio = 0.22).

## 3. Discussion

COPD, as a multifactorial disease, is characterized by complex gene–gene and gene–environment interactions, which are, until now, not completely well understood [35,36,37,38,39]. Indeed, many genes are associated with COPD genesis and development, as well as to its therapy [35,36,37,38,39]. Genetic variants (SNPs) are associated with COPD susceptibility [35,36,37,38,39], COPD severity (i.e., p53, [40]), COPD airway hyperreactivity [40], FEV_1_ [41], and therapeutic response [20,21,37,42,43].

In this study, we showed that FKBP5 rs4713916 is associated with a better response to rehabilitation in Italian patients with COPD. Thus, in the GA carrier patients, the 6MWD increased by 152.0 ± 26.95 m, whereas in the GG patients it increased by 89.87 ± 82.31 m (*p* = 0.0211). Moreover, in the GA patients there was a higher tendency in lung function improvement, such as dyspnea decrease (Borg scale), and in health status (SGRQ total score). In the GA patients, the level of circulating serum bilirubin was very high (0.94 ± 0.55 mg/dL), higher than in the GG patients (0.58 ± 0.28 mg/dL). Literature data associates relatively high levels of bilirubin with a lower risk of respiratory disease and all-cause mortality [32,33,34]. Specifically, a large study, conducted on 504,206 adults with a median follow-up time of 8 years, showed that higher serum bilirubin concentrations were associated with a significantly lower incidence of COPD. Men with a concentration of bilirubin of 0.94 mg/dL were associated with an incidence rate (IR) per 10,000 person-years (95% CI) equal to 9.9 (8.5–11.5) (women equal to 7.5 (6.5–8.7)). A concentration of bilirubin of 0.58 mg/dL was associated with an IR equal to 14.4 (12.7–16.2) for men and 9.9 (8.7–11.1) for women [32]. A recent work [44] shows that higher circulating bilirubin concentrations are associated with a lower risk of AECOPD.

This study is carried out in a real-life setting. According to the European Working Group on relative effectiveness, real life trials are means to analyze medical data collected under real life conditions (i.e., how treatments/interventions are administered in everyday clinical practice) [45]. This qualifying characteristic may lead to discrepancies with the results obtained by randomized controlled trials for a given endpoints. For these reasons our patients were stratified according to 6MWT.

The 6MWT is an inexpensive and reproducible method to assess exercise tolerance and to predict mortality in COPD. A recent work concluded that the 6MWT can be conducted both in observational and interventional studies with similar accuracy and predictive outcomes [46,47,48]. A 6MWD less than 350 m progressively increases the risk of death and hospitalizations, and can be used to stratify patients to be included in studies aimed at modifying respiratory outcomes. The 6MWT protocol is standardized, and 6MWD changes have been associated to treatment effect. The 6MWD represents a clinically informative measure of patient status. There is still a debate to define the threshold in 6MWT as “positive” or “indicative of efficacy.” Several studies have validated a minimal clinically importance distance of 30 m in COPD and other illness conditions [46,47,48]; consequently, a response of this magnitude could be considered as an improvement in outcome. Yet, the responsiveness of 6MWD to pharmacologic interventions in COPD remains poorly studied and the value of the 6MWD as an outcome in clinical trials has been questioned [49]. Recently, Celli et al. [50] reported the 6MWT is not a valuable test to evaluate a treatment response to bronchodilators. In their study, patients under bronchodilators showed a Δ of 17.9 m after one month of treatment, while the Δ of the placebo group was of 48.7 m. In the above study, all patients walked for 359.7 m (bronchodilators arm) or 361.2 m (placebo arm) at the beginning of the rehabilitation program. Our patients walked 96.8 m at the beginning of the PR program, suggesting a more COPD severe conditions. Nevertheless, after PR they gained 93.56 m, which is consistent with the improvements expected in patients with these conditions. Moreover, all of our patients received ICS therapy, whereas, as shown in the Celli et al. meta-analysis, patients received corticosteroids in only one [50] of the analyzed studies (i.e., Budesonide, see Table 1, Section “Clinical Trials” of the supplement, reporting the work of Sharafhaneh et al.) [51].

### Strengths and Limitations

Although findings from this study clearly emphasize the potential of FKBP5 genotyping as a predictor of the functional response to PR, the number of patients analyzed was relatively small for a genetic study to explain the phenotypic differences, so replication, fully-powered, and association studies are required. Moreover, this one is a pilot study. Indeed, the sample size of the study was evaluated according to the algorithm proposed by Viechtbauer et al. for pilot studies [52]. To provide a preliminary evidence referring to a polymorphism, with an allelic frequency of 5%, it is recommended to collect a minimum of 59 subjects (*n* = Ln(1-γ)/Ln(1-π)) (i.e., (*n* = Ln(1.0.95)/(Ln(1-0.05))). To take into account the loss of samples of subjects due to technical problems, we raised this number to 71 [52].

Another limitation is the severity of our patients, which implies that they received pharmacological treatment, including CSs, for a long time before the admission to PRU. Although this consideration may limit the validity of our results, it should be considered that our study group was representative of the patients commonly admitted to PRU, and; therefore, these results can be considered quite informative. 

Another limitation is that, according to GOLD guidelines [7], our patients received CSs in combination with two bronchodilators (Salbutamol and Ipratropium bromide), thus the CSs therapeutic response might be influenced by the bronchodilators in some degree. However, this is a common condition also in previous studies performed on COPD patients, who received various combination of therapies, of which the combined effect is difficult to predict [53,54,55]. Finally, all patients show different comorbidities (cardiovascular, renal, diabetes, etc.) that may impact on disease course but, according to GOLD [7], are treated “per usual standard” regardless of COPD. Additionally, in this case the response to CSs may be influenced by other medications.

## 4. Methods

### 4.1. Subjects 

The study was approved by the San Raffaele Ethical Committee, and all participants gave consent to participate both in the clinical and genetic study (Prot 15/2013). It is an interventional non pharmacological study conducted in 71 consecutive patients, aged 70 years or older, suffering from GOLD 3–4 COPD [7], and admitted to the San Raffaele Pisana Pulmonary Rehabilitation Unit (PRU) between January 2013–December 2015 for a comprehensive 3-week PR program. All patients were part of a larger study on the application of Systems Medicine approaches [56], evaluated in real life with a multidisciplinary and multidimensional assessment.

### 4.2. Patient Characteristics Measured at Baseline And/Or after Completion of PR 

1. Demographics (age, gender, marital and employment status, educational); 2. medical history and life-style (i.e., smoking/alcohol habit, familiarity for COPD or additional diseases, comorbidities); 3. body mass index (BMI); 4. health-related QoL evaluated using the St. George’s Respiratory Questionnaire (SGRQ, scores range from 0 to 100) [57,58,59]; 5. QoL evaluated using the 36-item short form health survey (SF-36, scores range from 0 to 100). 6. Use of long term oxygen therapy (LTOT); 7. instrumental evaluation [spirometry, pulse oximetry, blood pressure, electrocardiography (ECG), heart beats]; 8. disease-specific respiratory status using the Medical Research Council (MRC) (scores range from 0 to 5), the Barthel and the Borg scales for assessing dyspnea (scores range from 0 to 10); and the Maugeri Foundation Respiratory Failure Questionnaire (MRF 26, scores range from 0 to 5); 9. functional exercise capacity using the six-minute walking test (6MWT) [60]; 10. cognitive alterations/disorders using the Mini-Mental State Examination (MMSE, range from 0 to 30) and the Montreal Cognitive Assessment (MoCA, range from 0 to 30); 11. pharmacologic therapy; and 12. hematology and serological tests. 

### 4.3. Drug Therapy 

All patients received daily ICS [beclomethasone dipropionate: C_28_H_37_ClO_7_] 0.4 mg/mL in combination with two bronchodilators, such as one SABA [i.e., salbutamol: C_13_H_21_NO_3_] and one SAMA [anti subtype M3, i.e., ipratropium bromide: C_20_H_30_BrNO_3_] for 3 weeks. Exacerbations were treated with oral CSs prednisone [C_21_H_26_O_5_] at 5 or 25 mg, or betamethasone [C_22_H_29_FO_5_] at 1.5 mg or with injectable methylprednisolone [C_22_H_30_O_5_] at 20 mg.

### 4.4. Pulmonary Rehabilitation 

According to the national [53] and local (Lazio Country-Region) guidelines [54], patients with diagnosis of COPD received three weeks of hospitalized PR. The PR follows the Official American Thoracic Society/European Respiratory Society recommendations [55], with the final objective to improve both the physical and the psychological conditions and to promote the long-term adherence to health-enhancing behaviors [61].

### 4.5. DNA Extraction and Genotyping 

An aliquot of blood was taken from the sample used for routine hematological assays before and after rehabilitation and stored at −80 °C. 

All participants were genotyped for the rs6190, rs6189, rs41423247, rs4713916, rs37972, and rs2032582 polymorphisms. 

Genomic DNA was extracted from 2 mL peripheral EDTA-anticoagulated blood according to the manufacturer’s protocol (QIAamp DNA Blood Mini Kit; Qiagen Hilden, Germany). 

Specific polymorphisms were detected using MBK_MDR-1, MBK_FKBP5, MBK_GLCCI1, MBK_NR3C1/BclI, MBK_NR3C1/89, MBK_NR3C1/90 kits from Diatheva SrL (Fano, PU, Italy) according to company instructions. Allele-specific reaction mixtures were used to amplify 80 ng of genomic DNA. Cycling conditions were as follows: one step of 10 min at 95 °C, 40 cycles of 10 s at 95 °C (denaturation), touchdown from 62 °C to 59 °C (−1 °C each cycle for first 4 cycles) for 1 min (annealing and extension), and acquisition on FAM channel during annealing and extension step. Each allele was detected by a specific reaction mixture. Heterozygous samples showed the amplification of both alleles.

Melting profile of the allele variants for each polymorphism is reported in Appendix A (Appendix A).

### 4.6. Statistical Analysis

Data were managed and analyzed using STATA 12, and GraphPad Prism 8.1 (GraphPad Software Inc., La Jolla, CA, USA). Continuous variables were logarithmically transformed to normalize the distribution. Difference in the distribution of studied endpoints according to genotype were tested using *t* test, non-parametric Mann–Whitney U test, Kruskal–Wallis test; and Jonckheere–Terpstra trend test. 

To study the correlations between the variables, Spearman tests were used for Δ6MWT, gender, years of education, and FKBP5, where only FKBP5 was correlated negatively (−0.257, *p* = 0.034).

Pearson’s Chi Square tests were used for FKBP5 and GLCCl1; we could not reject the null hypothesis that FKBP5 and GLCCl1 were unrelated (χ^2^ = 7.45, *p* > 0.05, if the *p*-value is greater than alpha we can conclude that they are independent).

Multiple regression with predictor class of age (50–65; 65–75; 76–90) and FKBP5 (GG, GA) confirmed the independent test that showed that who owns the mutation GA’s FKBP5 walks an average of 81 m (SE 28.8) more than who has the mutation GG’s FKBP5, in the class of age range 76–90 (*p* = 0.009).

Data were described as mean ± SD (standard deviation). A *p*-value ≤ of 0.05 was considered as statistically significant.

## 5. Conclusions

This study shows, for the first time, the association between the rs4713916 polymorphism of the FKBP5 gene and different endpoints of IPR response in COPD patients. Specifically, GA carriers show better performances in the 6MWT and in lung functions.

## Figures and Tables

**Figure 1 ijms-20-02024-f001:**
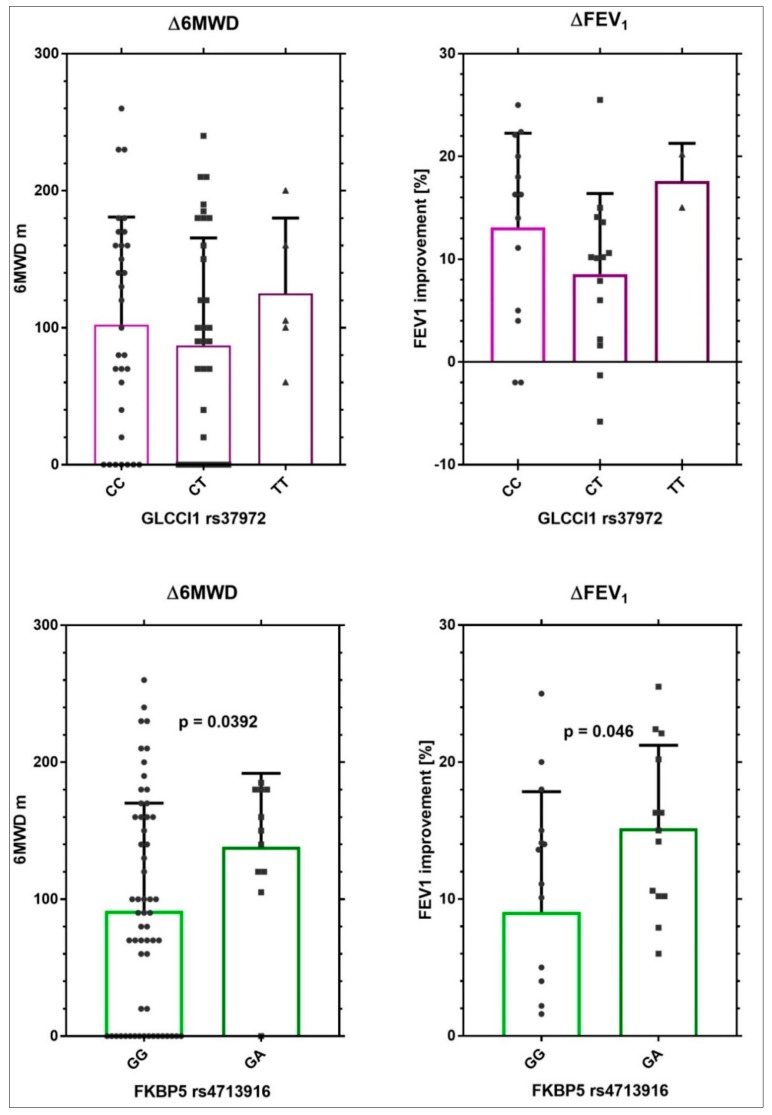
Improvement in Δ6MWD and FEV_1_ in patients stratified for FKBP5 rs4713916 or for GLCCI1 rs37972.

**Table 1 ijms-20-02024-t001:** Demographic and clinical characteristics of the analyzed sample of patients.

Variables	Patients *n* = 71
Males	30 (42.2%)
Females	41 (57.8%)
Years of Education	8.9 ± 4.08
Marital Status	
Single	3 (4.2%)
Married	36 (50.7%)
Divorced/widow	32 (45.1%)
No smokers	6 (8.5%)
Current smokers	11 (15.5%)
Ex-smokers	46 (64.7%)
Not Responders	8 (11.3%)
Occupational Status	
Retired	63 (88.7%)
Housewife	8 (11.3%)
BMI	27.57 ± 4.9
Therapy With O_2_	23 (32.4%)
Corticosteroids therapy	71 (100%)
MRC dyspnea grade	4.0 ± 0
Borg grade	7.87 ± 0.92
SGRQ-Total points	49.33± 16.02
6MWD (meters)	96.76 ± 85.25
MRF26	72.14 ± 15.65
Barthel	68.32 ± 24.37
FEV_1_	48.40 ± 24.2
MMSE	26.78 ± 2.83
MoCa	25.57 ± 3.73
SF-36 General Health	73.40 ± 11.54
SF-36 Mental Health	62.56 ± 8.54
CIRS-severity	1.58 ± 0.23
CIRS-comorbidity	2.44 ± 1.36

**Table 2 ijms-20-02024-t002:** Respiratory parameters before and after pulmonary rehabilitation (PR).

	Before PR (x ± SD)	After PR (x ± SD)	Δ	*p* Value *
6MWD (meters)	96.76 ± 85.25	191.1 ± 132.7	94.37 ± 79.97	<0.0001
FEV_1_ (%)	48.11 ± 23.35	62.46 ± 11.74	11.32 ± 8.42	0.0073
MRC	4.0 ± 0.0	3.34 ± 0.61	−0.66 ±0.61	<0.0001
Borg	7.87 ± 0.92	5.20 ± 2.47	2.676 ± 1.911	<0.0001
SGRQ	49.33 ± 16.02	44.29 ± 16.08	−2.59 ± 16.69	NS
Barthel	68.32 ± 24.37	86.1 ± 16.72	17.77 ± 15.01	<0.0001
MRF26	72.14 ± 15.65	48.69 ± 19.07	−23.3 ± 14.17	<0.0001

* According to non-parametric Mann–Whitney test.

**Table 3 ijms-20-02024-t003:** Allele frequencies of single-nucleotide polymorphisms among four gene in COPD patients.

GLCCI1 rs37972	NR3C1 rs6189	NR3C1 rs6190	NR3C1 Bcl2 rs41423247	MDR-1 rs2032582	FKBP5 rs4713916
CC 31 (43.7%)	GG 66 (93%)	GG 66 (93%)	GG 35 (49.3%)	GG 25 (35.2%)	GG 57 (80.3%)
CT 34 (47.9%)	GA 3 (4.2%)	GA 3 (4.2%)	GC 28 (39.4%)	GA 3 (4.2%)	GA 13 (18.3%)
TT 5 (7%)	-	-	CC 7 (9.9%)	GT 22 (31%)	-
-	-	-	-	TT 20 (28.2%)	-
NA 1 (1.4%)	NA 2 (2.8%)	NA 2 (2.8%)	NA 1 (1.4%)	NA 1 (1.4%)	NA 1 (1.4%)

NA—not amplified; rs37972, —intron variant, upstream variant 2KB [25]; rs6189, —missense, synonymous codon, UTR variant 5 prime [26]; rs6190—missense, UTR variant 5 prime [27]; rs41423247—intron variant [28]; rs2032582—missense [29]; rs4713916—intron variant [30].

**Table 4 ijms-20-02024-t004:** Genotype and allele distribution of single-nucleotide polymorphisms among the four genes in COPD patients and response to ICS according to 6MWD. Δ > 50 m = responders, Δ < 30 m = non-responders.

SNP	Responders Patients Number (% Over Total)	Non-Responders Patients Number (% Over Total)	% Responders Genotype	*p* *
rs37972				
*CC*	22 (32.35%)	8 (11.77%)	73.3%	0.08
*CT*	21(30.88%)	12 (17.65%)	62.6%	
*TT*	5 (7.35%)	0	100%	
rs4713916				
*GG*	37 (55.22%)	19 (28.36%)	60%	0.039
*GA*	10 (14.92%)	1 (1.5%)	90.91%	
rs6189				
*GG*	45 (67.16%)	19 (28.35%)	70.31%	NS
*GA*	3 (4.48%)	0	100%	
rs6190				
*GG*	44 (65.67%)	20 (29.85%)	68.75%	NS
*GA*	3 (4.48%)	0	100%	
rs41423247				
*CC*	4 (5.88%)	3 (4.41%)	57.14%	NS
*GC*	19 (27.94%)	7 (10.29%)	73.07%	
*GG*	25 (36.76%)	10 (14.71%)	71.42%	
rs2032582				
*GG*	14 (20.59%)	10 (14.71%)	58.33%	NS
*GA*	3 (4.41%)	0	100%	
*GT*	15 (22.06%)	6 (8.82%)	71.43%	
*TT*	16 (23.53%)	4 (5.88%)	80%	

* According to Mann–Whitney test.

**Table 5 ijms-20-02024-t005:** Associations between FKBP5 variant rs47139 and response to PR in COPD patients.

Characteristics	All GA	All GG	*p* Value
Δ Borg	−3.50 ± 1.35	−2.35 ± 1.99	0.0862
Δ SGRQ-Total points	14.0 ± 24.17	−4.14 ± 10.68	0.037
Δ Basophiles *	0.15 ± 0.28	−0.11 ± 0.26	0.085
Bilirubin **	0.94 ± 0.55	0.58 ± 0.29	0.0453

* percentage × 10^−3^, ** mg/dL.

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
