# Peer review of "FKBP5 rs4713916: A Potential Genetic Predictor of Interindividual Different Response to Inhaled Corticosteroids in Patients with Chronic Obstructive Pulmonary Disease in a Real-Life Setting"

_ijms, 2019, doi:10.3390/ijms20082024_

Round 1

Reviewer 1 Report

In this manuscript, Russo and colleagues evaluated potential relation of the single nucleotide polymorphisms (SNPs) of NR3C1, GLCCI1, FKBP5 and MDR1 to the different response to corticosteroids in COPD patients. Interestingly, authors found FKBP5 gene (rs4713916) has a significant association with functional outcomes in COPD patients.

This novel work might be helpful to provide a precision medicine approach to better treating COPD. Results are convincing and illustrated satisfactorily.

However, methods section should include more experimental details (e.g. primer sequence used in all the experiments).

Author Response

"However, methods section should include more experimental details (e.g. primer sequence used in all the experiments)."

In the revised manuscript we included more experimental details:

Specific polymorphisms were detected using MBK_MDR-1, MBK_FKBP5, MBK_GLCCI1, MBK_NR3C1 / BclI, MBK_NR3C1 / 89, MBK_NR3C1 / 90 kits from Diatheva SrL (Italy) according to company instructions. Allele-specif reactions mixtures were used to amplify 80 ng of genomic DNA. Cycling conditions were as follows: one step of 10 min at 95°C, 40 cycles of 10 s at 95°C (denaturation), touchdown from 62°C to 59°C (-1°C each cycle for first 4 cycles) 1 min (annealing and extension), acquisition on FAM channel during annealing and extension step. Each allele is detected by a specific reaction mixture. Heterozyous samples showed the amplification of both allele.

Reviewer 2 Report

The results of the study are of interest, but the number of participants is an important limitation of the work. I recommend extending the study to a greater number of participants to have generalized results.

Author Response

"The results of the study are of interest, but the number of participants is an important limitation of the work. I recommend extending the study to a greater number of participants to have generalized results".

This one is a pilot study. Indeed, the sample size of the study has been evaluated according to the algorithm proposed by Viechtbauer et al for pilot studies [2015]. To provide a preliminary evidence referring to a polymorphism with an allelic frequency of 5% it is recommended to collect a minimum of 59 subjects (n=Ln(1-γ)/Ln(1-π)), i.e., (n=Ln(1.0.95)/(Ln(1-0.05)). To take into account the loss of samples of subjects due to technical problems we raised this number to 71 [Viechtbauer et al, 2015].

Viechtbauer, W.; Smits, L.; Kotz, D.; Budé, L.; Spigt, M.; Serroyen, J.;Crutzen, R. A simple formula for the calculation of sample size in pilot studies. J. Clin Epidemiol. 2015, 68, 1375-1379.

We planned to extend our study to a large population, however we could have significant results in one year.

Round 2

Reviewer 2 Report

The comments of the reviewers have been answered in an appropriate manner.